

# Entanglement classification via operator size

**Qi-Feng Wu***

Department of Physics and Center for Field Theory and Particle Physics,
Fudan University, Shanghai, 200433, China
Yau Mathematical Sciences Center, Tsinghua University, Beijing 100084, China
Department of Physics and Astronomy, Ghent University, 9000 Ghent, Belgium

* qifeng.wu@ugent.be

## Abstract

In this work, multipartite entanglement is classified by polynomials. I show that the operator size is closely related to the entanglement structure. Given a generic quantum state, I define a series of subspaces generated by operators of different sizes acting on it. The information about the entanglement is encoded into these subspaces. With the dimension of these subspaces as coefficients, I define a polynomial which I call the entanglement polynomial. The entanglement polynomial induces a homomorphism from quantum states to polynomials. It implies that we can characterize and find the building blocks of entanglement by polynomial factorization. Two states share the same entanglement polynomial if they are equivalent under the stochastic local operations and classical communication. To calculate the entanglement polynomial practically, I construct a series of states, called renormalized states, whose ranks are related to the coefficients of the entanglement polynomial.

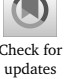

# 1  Introduction

Operator size is a measure of local degrees of freedom on which an operator has a nontrivial action [1, 2]. It is also a characteristic of operator scrambling [3, 4]. In AdS/CFT, it was conjectured that the size of a boundary operator is related to the bulk radial momentum [5–7]. However, there is an ambiguity that operators of different sizes can map a given state to another same state, which can be understood in terms of stabilizer formalism [8, 9]. A similar issue occurs in AdS/CFT due to its quantum error correction property [10]. To fix such an ambiguity, I generalize the operator size to be state-dependent, such that operators have the same size if they have the same action on a given state [11]. For a given state $|\psi\rangle$, the state-dependent operator size of an operator $\mathcal{O}$ can be defined as the expectation value of a state-dependent operator $n_{|\psi\rangle}$ called size operator,

$$\mathcal{S}_{|\psi\rangle}(\mathcal{O}) \equiv \frac{\langle\psi|\mathcal{O}^\dagger n_{|\psi\rangle}\mathcal{O}|\psi\rangle}{\langle\psi|\mathcal{O}^\dagger\mathcal{O}|\psi\rangle} \, . \tag{1}$$

A similar definition of state-dependent operator size is proposed in [12].

In this paper, I will not discuss the entanglement dynamics mentioned above but focus on the classification of entanglement structures. Entanglement classification is a long-standing open problem in quantum information theory. Various methods were proposed [13–20], but our understanding of multipartite entanglement is still limited. I show that the entanglement structure of $|\psi\rangle$ is encoded into eigenspaces of $n_{|\psi\rangle}$. As we will see, the dimensions of these eigenspaces are characteristics of entanglement. They form the coefficients of a polynomial which can classify multipartite entanglement. So I call it the entanglement polynomial.

I prove that the entanglement polynomial is a SLOCC invariant and thus is a characteristic of the entanglement [13]. Here the SLOCC stands for stochastic local operations and classical communication. As a map from quantum states to polynomials, the entanglement polynomial preserves the product structure, which means the entanglement polynomial of the tensor product of two states is the product of the entanglement polynomials of these two states. In other words, entanglement polynomial induces a homomorphism. More precisely, we will see that it is a monoid homomorphism. Since product states are mapped to reducible polynomials, we can find and characterize the building blocks of entanglement by polynomial factorization.

We can construct an isomorphism by taking the quotient over the kernel of homomorphism. Applying this fact to the entanglement polynomial, we obtain an isomorphism from equivalence classes of states to polynomials. Since entanglement polynomial is a SLOCC invariant, states of the same equivalence class share a similar entanglement structure. Thus, the entanglement polynomial induces a classification of entanglement. It can be shown that the number of these classes is finite if the Hilbert space is finite-dimensional.

This paper is organized as follows. In Section 2, we introduce some relevant concepts and define the entanglement polynomial. In Section 3, we show that the entanglement polyno-

mial induces a homomorphism from quantum states to polynomials. By taking the quotient over the kernel of this homomorphism, we obtain an isomorphism from entanglement classes to polynomials. In Section 4, we show that there exists a class of operators preserving the entanglement polynomial of any state. Two examples of such operators are local invertible operators and permutation operators. We call these operators the symmetries of the entanglement polynomial. In Section 5, we construct a series of states called renormalized states. We show that the ranks of these states are related to the coefficients of entanglement polynomials. We can use this relation to calculate the entanglement polynomial practically. In Section 6, we summarize the results of this paper. In Appendix A, we prove that entanglement polynomial preserves the product structure. In Appendix B, we illustrate how the factorization of the entanglement polynomials is related to the size additivity. In Appendix C, a theorem about the symmetry of the entanglement polynomial is proven. In Appendix D, the relation between renormalized states and coefficients of the entanglement polynomial is proven.

## 2 Entanglement polynomial

We first review several relevant notions and then define the entanglement polynomial. For simplicity, we consider quidts.

The Hilbert space of $N$ qudits is given by

$$\mathcal{H} = \bigotimes_{i=1}^{N} \mathcal{H}_i, \qquad \mathcal{H}_i \cong \mathbb{C}^d. \tag{2}$$

$\mathcal{H}_i$ is the Hilbert space of the $i$-th qudit. Suppose that $B$ is a subset of these qudits and $|B|$ is the number of qudits of it. We denote the linear space of operators acting on $B$ nontrivially as

$$\mathcal{A}_B = \left\{ \mathcal{O}_B \otimes I_{\bar{B}} \,|\, \mathcal{O}_B \in \text{End}(\mathcal{H}_B), \ \mathcal{H}_B = \bigotimes_{i \in B} \mathcal{H}_i \right\}. \tag{3}$$

$\bar{B}$ denotes the complement of $B$. $I_{\bar{B}}$ is the identity acting on $\mathcal{H}_{\bar{B}}$. Then we define a linear space of operators as

$$V^k \equiv \sum_{|B|=k} \mathcal{A}_B. \tag{4}$$

The summation here means that $V^k$ is spanned by the elements in $\mathcal{A}_B$'s. We set $V^0 \equiv \text{Span}\{I\}$. Operators in $V^k$ are called $k$-local operators. We can see that a $k$-local operator can always be written as a linear combination of the tensor products of $k$ local operators. Given a state $|\psi\rangle$, we can construct the subspace

$$W_k \equiv \text{Span}\{\mathcal{O}|\psi\rangle \,\big|\, \mathcal{O} \in V^k\}. \tag{5}$$

$W_k$ can be decomposed as follows

$$W_k = \Delta W_k \oplus W_{k-1}, \quad \Delta W_k \perp W_{k-1}, \quad 0 \leq k \leq N_{|\psi\rangle}. \tag{6}$$

$N_{|\psi\rangle}$ is the minimal value of $k$ such that $W_{k+1} = W_k$. We set $W_{-1}$ as the empty set. The meaning of $\Delta W_k$ is that we only need $k$-local operators acting on $|\psi\rangle$ to generate $\Delta W_k$ which cannot be generated by $(k-1)$-local operators. Each state in $\Delta W_k$ can be perfectly distinguished from states that can be generated by $(k-1)$-local operators acting on $|\psi\rangle$. Then we can say that there are $k$ local d.o.f. that are changed when $|\psi\rangle$ evolves to a state in $\Delta W_k$. For this reason, we call $\Delta W_k$ the $k$-local subspace. In [11], the eigenspaces of the size operator in Eq. (1) are defined as the $k$-local subspaces. For more details, see Appendix B.

Then we define the entanglement polynomial as

$$f(|\psi\rangle) \equiv \sum_{k=0}^{N_{|\psi\rangle}} |\Delta W_k| x^k. \tag{7}$$

$|\Delta W_k|$ is the dimension of $\Delta W_k$. $x$ is a symbol. This polynomial encodes the information about the number of states that can be generated by $k$-local operators, which is a property closely related to the entanglement structure.

## 3 Isomorphism

A monoid is a set that is closed under an associative binary operation and has an identity element. As mentioned in the introduction, the entanglement polynomial induces a monoid homomorphism from states to polynomials.

To illustrate it, we consider an infinite number of qudits in this section. Denote the set of quantum states of all of the subsystems of these qudits as $S$. We can show that $S$ is a monoid with the tensor product as the associative binary operation. To check it, we first note that the tensor product between two quantum states is a binary operation and it is associative. Since two non-overlapping subsystems still form a subsystem, $S$ is closed under the tensor product. To specify the identity, we note that the empty set $\emptyset$ is also a subsystem of qudits, and the union of $\emptyset$ and an arbitrary subsystem $B$ is still $B$. So the quantum state of $\emptyset$ is the identity w.r.t. the tensor product. We denote it as $|\emptyset\rangle$. Note that the Hilbert space dimension of $n$ qudits is $d^n$. Since $\emptyset$ is the set of zero qudits, its Hilbert space is one-dimensional. So $|\emptyset\rangle$ is unique. Then we have

$$|\psi\rangle \otimes |\emptyset\rangle = |\emptyset\rangle \otimes |\psi\rangle = |\psi\rangle, \quad |\psi\rangle, |\emptyset\rangle \in S. \tag{8}$$

Putting these facts together, we can see that $S$ is indeed a monoid.

The entanglement polynomial maps $S$ to a set of polynomials denoted by $f(S)$. $f(S)$ is also a monoid. To see this, we note that no nontrivial operator acts on the Hilbert space of zero qudits, so we have

$$f(|\emptyset\rangle) = 1. \tag{9}$$

Thus $f(|\emptyset\rangle)$ is the identity w.r.t. the polynomial multiplication " $\cdot$ ". Obviously, " $\cdot$ " is an associative binary operation. In Appendix A, I prove that

$$f(|\psi\rangle \otimes |\phi\rangle) = f(|\psi\rangle) \cdot f(|\phi\rangle), \tag{10}$$

for arbitrary two states. Thus, $f(S)$ is closed under " $\cdot$ ". Collecting these facts, we can see that $f(S)$ is also a monoid. Eq. (10) also implies that $f$ preserves the product structure, so we have the following theorem.

**Theorem 1** *The entanglement polynomial induces a monoid homomorphism from the state monoid $S$ to the polynomial monoid $f(S)$.*

We can construct an isomorphism by taking quotient over the kernel of the homomorphism. To take the quotient, we define an equivalence relation "$\sim$" over $S$ such that $|\psi\rangle \sim |\phi\rangle$ if $f(|\psi\rangle) = f(|\phi\rangle)$. Obviously, we have

$$|\psi\rangle \sim |\psi\rangle, \tag{11a}$$

$$|\psi\rangle \sim |\phi\rangle \Rightarrow |\phi\rangle \sim |\psi\rangle, \tag{11b}$$

$$|\psi\rangle \sim |\phi\rangle, |\phi\rangle \sim |\chi\rangle \Rightarrow |\psi\rangle \sim |\chi\rangle. \tag{11c}$$

So "$\sim$" is indeed an equivalence relation. In fact, it is a congruence relation, which means that

$$|\psi_1\rangle \sim |\phi_1\rangle, \; |\psi_2\rangle \sim |\phi_2\rangle \Rightarrow |\psi_1\rangle \otimes |\psi_2\rangle \sim |\phi_1\rangle \otimes |\phi_2\rangle \tag{12}$$

is further satisfied. The equivalence class of $|\psi\rangle$ under "$\sim$" is denoted by $[|\psi\rangle]$. Since "$\sim$" is a congruence relation, the "tensor product" between $[|\psi\rangle]$ and $[|\phi\rangle]$ can be uniquely defined by

$$[|\psi\rangle] \otimes [|\phi\rangle] \equiv [|\psi\rangle \otimes |\phi\rangle] \,. \tag{13}$$

Now we can define the quotient monoid $S/\ker f$ by

$$S/\ker f \equiv \{[|\psi\rangle] \,\big|\, |\psi\rangle \in S\} \,, \tag{14}$$

with the tensor product defined in Eq. (13) as the associative binary operation. According to the first isomorphism theorem, the polynomial monoid $f(S)$ is isomorphic to $S/\ker f$,

$$f(S) \cong S/\ker f \,. \tag{15}$$

To specify the isomorphism map, we define the entanglement polynomial of an equivalence class by

$$f([|\psi\rangle]) \equiv f(|\psi\rangle) \,. \tag{16}$$

Since equivalent states have the same entanglement polynomial, this definition is unique. Then we have

**Theorem 2** *The entanglement polynomial induces an isomorphism from the quotient monoid $S/\ker f$ to the polynomial monoid $f(S)$.*

According to Eq. (10), product states are mapped to reducible polynomials in $f(S)$, so we can find basic building blocks of entanglement by polynomial factorization. For this reason, we call elements in $S/\ker f$ the entanglement classes.

## 4 Symmetry

Besides calculating by definition, symmetries of entanglement polynomials can determine whether two states are in the same entanglement class. By symmetry, I mean an operator $g$ satisfying

$$f(g|\psi\rangle) = f(|\psi\rangle) \,. \tag{17}$$

Unlike the case of groups, the kernel of a monoid homomorphism cannot be determined by the preimage of the identity. So the symmetry generally depends on the specific state $|\psi\rangle$. However, we can still find some state-independent symmetries. Define the adjoint action of an invertible operator $g$ by

$$\mathrm{ad}_g \mathcal{O} \equiv g^{-1} \mathcal{O} g \,. \tag{18}$$

The adjoint action on an operator space $M$ is defined by

$$\mathrm{ad}_g M \equiv \{\mathrm{ad}_g \mathcal{O} \,\big|\, \mathcal{O} \in M\} \,. \tag{19}$$

Then we have the following theorem (for the proof, see Appendix C).

**Theorem 3** *If $\text{ad}_g$ induces an automorphism on the space of 1-local operators $V$, then $g$ keeps entanglement polynomials invariant, i.e.*

$$\text{ad}_g V = V \Rightarrow f(g|\psi\rangle) = f(|\psi\rangle), \quad \forall \, |\psi\rangle. \tag{20}$$

As an explicit example, we show that if two states are equivalent under stochastic local operations and classical communication (SLOCC), they are in the same entanglement class. In [13], it was proven that $|\psi\rangle$ and $|\phi\rangle$ are equivalent under SLOCC if and only if there exists an invertible local operator $L$ such that

$$|\psi\rangle = L|\phi\rangle. \tag{21}$$

Given a set of $N$ qudits, an invertible local operator $L$ is an operator of the following form

$$L = \bigotimes_{i=1}^{N} L_i, \quad L_i \in \text{Aut}(\mathcal{H}_i). \tag{22}$$

$L_i$ is an invertible operator acting on the $i$-th qudit. By definition, operators in $V^k$ can be expressed as

$$\mathcal{O} = \sum_{|B|=k} c_B I_{\bar{B}} \otimes \bigotimes_{i \in B} \mathcal{O}_i. \tag{23}$$

$B$ labels subsets of $N$ qudits. $|B|$ is the number of qudits in $B$. $\mathcal{O}_i \in \text{End}(\mathcal{H}_i)$, $c_B \in \mathbb{C}$. Then we have

$$\text{ad}_L \mathcal{O} \equiv L^{-1} \mathcal{O} L = \sum_{|B|=k} c_B I_{\bar{B}} \otimes \bigotimes_{i \in B} L_i^{-1} \mathcal{O}_i L_i. \tag{24}$$

Note that $L_i^{-1} \mathcal{O}_i L_i \in \text{End}(\mathcal{H}_i)$ and $L$ is invertible, so $\text{ad}_L$ induces an automorphism on $V^k$ for all $k$. Thus, Theorem 3 implies the following theorem.

**Theorem 4** *If two states are equivalent under SLOCC, then their entanglement polynomials are equal.*

Another example of the symmetry is the permutation. A permutation can be generated by a series of swap operators $X_i$ satisfying

$$X_i|a\rangle_i|b\rangle_{i+1} = |b\rangle_i|a\rangle_{i+1}, \tag{25}$$

for all $|a\rangle_i, |b\rangle_i \in \mathcal{H}_i$ and $|a\rangle_{i+1}, |b\rangle_{i+1} \in \mathcal{H}_{i+1}$. Then the action of $\text{ad}_{X_i}$ on $\mathcal{O}$ is to swap $\mathcal{O}_i$ and $\mathcal{O}_{i+1}$, such that $X_i^{-1} \mathcal{O} X_i \in V^k$. Since $X_i$ is invertible, $\text{ad}_{X_i}$ induces an automorphism. So swap operators are symmetries of entanglement polynomials. Since each permutation in the permutation group $\text{Sym}_N$ is the product of a series of $X_i$'s, it also keeps the entanglement polynomials invariant. So two states in the same SLOCC class or $\text{Sym}_N$ class must belong to the same entanglement class.

According to Eq. (5), the coefficients of an entanglement polynomial is an integer partition of the Hilbert space dimension, which means it can always classify states into a finite number of entanglement classes if the Hilbert space is finite-dimensional.

## 5 Renormalized state

To calculate the entanglement polynomial efficiently, we construct a series of states whose ranks are related to entanglement polynomials. Given a state $\rho$, we define the renormalized states as

$$\rho_k \equiv \frac{\sum_{|B|=k} \rho_{\bar{B}}}{\sum_{|B|=k} \text{tr}(\rho_{\bar{B}})}. \tag{26}$$

Table 1: Entanglement polynomials of genuinely entangled states up to three qubits.

| $\lvert\psi\rangle$ | $\lvert 0\rangle$ | $\lvert\text{Bell}\rangle$ | $\lvert\text{W}\rangle$ | $\lvert\text{GHZ}\rangle$ |
|---|---|---|---|---|
| $f(\lvert\psi\rangle)$ | $1+x$ | $1+3x$ | $1+6x+x^2$ | $1+7x$ |

Table 2: Entanglement polynomials of product states up to three qubits.

| $\lvert\psi\rangle$ | $\lvert 00\rangle$ | $\lvert 000\rangle$ | $\lvert 0\rangle\lvert\text{Bell}\rangle$ |
|---|---|---|---|
| $f(\lvert\psi\rangle)$ | $(1+x)^2$ | $(1+x)^3$ | $(1+x)(1+3x)$ |

$\rho_{\bar{B}} \in \mathcal{A}_{\bar{B}}$ is the reduced density matrix satisfying

$$\text{tr}\left(\mathcal{O}_{\bar{B}}\rho\right) = \text{tr}\left(\mathcal{O}_{\bar{B}}\rho_{\bar{B}}\right), \quad \forall\, \mathcal{O}_{\bar{B}} \in \mathcal{A}_{\bar{B}}. \tag{27}$$

To illustrate the physical meaning of $\rho_k$, suppose the system is in a state $\rho$, and we try to measure the operator $\mathcal{O}$ in a noisy laboratory. The effect of the noise is to disturb our observation such that we cannot access some parts of the system. The noise is stochastic, so the position of the lost part $B$ is random. At each observation, the expectation value of $\mathcal{O}$ is

$$\langle\mathcal{O}\rangle_{\text{noise}} = \text{tr}\left(\mathcal{O}\rho_{\bar{B}}\right). \tag{28}$$

If we perform many experiments, the average of the expectation value is[1]

$$\overline{\langle\mathcal{O}\rangle_{\text{noise}}} = \frac{\sum_B \text{tr}(\mathcal{O}\rho_{\bar{B}})}{\sum_B \text{tr}(\rho_{\bar{B}})}. \tag{29}$$

If the size of the lost part is always equal to $k$, then we have

$$\overline{\langle\mathcal{O}\rangle_{\text{noise}}} = \text{tr}(\mathcal{O}\rho_k). \tag{30}$$

So the renormalized states are effective states under the random noise of a given size.

Then we can show that, given a pure state $\rho = \lvert\psi\rangle\langle\psi\rvert$,

$$\lvert W_k\rvert = \text{Rank}(\rho_k). \tag{31}$$

See Appendix D for the proof. As an example, we apply Eq. (31) to qubits. According to Theorem 4, we only need to perform calculations for representatives of SLOCC classes and permutation classes. Up to three qubits, they are $\lvert 0\rangle$,

$$\lvert\text{Bell}\rangle = \frac{1}{\sqrt{2}}(\lvert 00\rangle + \lvert 11\rangle), \tag{32a}$$

$$\lvert\text{W}\rangle = \frac{1}{\sqrt{3}}(\lvert 100\rangle + \lvert 010\rangle + \lvert 001\rangle), \tag{32b}$$

$$\lvert\text{GHZ}\rangle = \frac{1}{\sqrt{2}}(\lvert 000\rangle + \lvert 111\rangle), \tag{32c}$$

and tensor products of them, i.e. $\lvert 00\rangle$, $\lvert 000\rangle$, $\lvert 0\rangle\lvert\text{Bell}\rangle$. The results are listed in Table 1 and Table 2. We can see that the entanglement polynomial clearly distinguishes the states of different entanglement structures.

---

[1]Here we suppose the probability distribution of the lost parts is flat. We can consider general distribution, see Appendix D.

## 6 Conclusion

We have defined the entanglement polynomial which induces a monoid isomorphism from entanglement classes to polynomials. The tensor product among states is mapped to the polynomial product, which implies that we can find the basic building blocks of entanglement by polynomial factorization. If an operator induces automorphisms on all the subspaces of $k$-local operators, then it keeps the entanglement polynomial invariant. As a consequence, the entanglement polynomial is proven to be a SLOCC invariant. We then construct the renormalized states and show that, by calculating their ranks, we can calculate the entanglement polynomial practically.

## Acknowledgements

I would like to thank Ling-Yan Hung, Xiao-Liang Qi, Yong-Shi Wu, Xin-Yang Yu, and Si-Nong Liu for their helpful discussions.

**Funding information**   I acknowledge the financial support of Fudan University, Tsinghua University and Ghent University.

## A   Proof of $f(|\psi\rangle \otimes |\phi\rangle) = f(|\psi\rangle) \cdot f(|\phi\rangle)$

For later convenience, we define the action of an operator space $M$ on a Hilbert space $\mathcal{H}$ by

$$M\mathcal{H} \equiv \mathrm{Span}\left\{\mathcal{O}|\psi\rangle \,\middle|\, \mathcal{O} \in M, |\psi\rangle \in \mathcal{H}\right\}. \tag{A.1}$$

Denote the space spanned by $|\psi\rangle$ as $\mathcal{H}^{|\psi\rangle}$. We can rewrite the subspace defined by Eq. (5) as

$$W_k^{|\psi\rangle} = V^k \mathcal{H}^{|\psi\rangle}. \tag{A.2}$$

Notice that we add a superscript to $W_k$ to indicate its dependence on $|\psi\rangle$. $V^k \mathcal{H}$ can be obtained by $V$ acting on $\mathcal{H}$ $k$ times, which is

$$V^k \mathcal{H} = (V)^k \mathcal{H} \equiv \underbrace{VV\ldots V}_{k}\mathcal{H}. \tag{A.3}$$

$V$ is the space of 1-local operators. We can also define the multiplication between two operator spaces $V_1$ and $V_2$

$$V_1 V_2 \equiv \mathrm{Span}\{\mathcal{O}_1 \mathcal{O}_2 \,\middle|\, \mathcal{O}_1 \in V_1, \mathcal{O}_2 \in V_2\}, \tag{A.4}$$

such that we have

$$V^k = (V)^k \equiv \underbrace{VV\ldots V}_{k}. \tag{A.5}$$

So we can rewrite $(V)^k$ as $V^k$ for simplicity. The symbol "$\Delta$" in Eq. (6) can be defined as an operator $\Delta_k$ acting the the Hilbert space with the index $k$, i.e.

$$W_k = \Delta_k W_k \oplus W_{k-1}, \quad \Delta_k W_k \perp W_{k-1}. \tag{A.6}$$

Suppose a system is divided into two parts, $B$ and $\bar{B}$. Denote the space of 1-local operators acting on $B$ and $\bar{B}$ as $V_B$ and $V_{\bar{B}}$ respectively. Then we have

$$V = V_{B \cup \bar{B}} = V_B + V_{\bar{B}}. \tag{A.7}$$

Take the $k$-th power

$$V^k = (V_B + V_{\bar{B}})^k = \sum_{l+m=k} (V_{\bar{B}})^m (V_B)^l . \tag{A.8}$$

Given a state $|\psi\rangle \otimes |\phi\rangle$ with $|\psi\rangle \in \mathcal{H}_B$ and $|\phi\rangle \in \mathcal{H}_{\bar{B}}$. Using Eq. (A.8), we have

$$
\begin{aligned}
V^k(\mathcal{H}^{|\psi\rangle} \otimes \mathcal{H}^{|\phi\rangle}) &= \sum_{l+m=k} (V_{\bar{B}})^m (V_B)^l \left( \mathcal{H}^{|\psi\rangle} \otimes \mathcal{H}^{|\phi\rangle} \right) \\
&= \sum_{l+m=k} V_{\bar{B}}^m V_B^l \left( \mathcal{H}^{|\psi\rangle} \otimes \mathcal{H}^{|\phi\rangle} \right) \\
&= \sum_{l+m=k} V_{\bar{B}}^m \bigoplus_{p=0}^{l} \Delta_p \left( V_B^p \left( \mathcal{H}^{|\psi\rangle} \otimes \mathcal{H}^{|\phi\rangle} \right) \right) \\
&= \sum_{l+m=k} V_{\bar{B}}^m \bigoplus_{p=0}^{l} \Delta_p (V_B^p \mathcal{H}^{|\psi\rangle}) \otimes \mathcal{H}^{|\phi\rangle} \\
&= \sum_{l+m=k} \bigoplus_{p=0}^{l} \Delta_p \left( V_B^p \mathcal{H}^{|\psi\rangle} \right) \otimes V_{\bar{B}}^m \mathcal{H}^{|\phi\rangle} \\
&= \sum_{l+m=k} \bigoplus_{p=0}^{l} \bigoplus_{q=0}^{m} \Delta_p \left( V_B^p \mathcal{H}^{|\psi\rangle} \right) \otimes \Delta_q \left( V_{\bar{B}}^q \mathcal{H}^{|\phi\rangle} \right) \\
&= \bigoplus_{l+m=k} \bigoplus_{p=0}^{l} \bigoplus_{q=0}^{m} \Delta_p \left( V_B^p \mathcal{H}^{|\psi\rangle} \right) \otimes \Delta_q \left( V_{\bar{B}}^q \mathcal{H}^{|\phi\rangle} \right) \\
&= \bigoplus_{l+m=k} \bigoplus_{p=0}^{l} \bigoplus_{q=0}^{m} \Delta_p W_p^{|\psi\rangle} \otimes \Delta_q W_q^{|\phi\rangle} .
\end{aligned}
\tag{A.9}
$$

In the first equality, we used Eq. (A.8). In the second equality, we rewrite $(V_{\bar{B}})^m$ and $(V_B)^l$ as $V_{\bar{B}}^m$ and $V_B^l$ respectively for simplicity. In the third equality, We used Eq. (A.6) to expand $V_B^l(\mathcal{H}^{|\psi\rangle} \otimes \mathcal{H}^{|\phi\rangle})$ in terms of $\Delta_p(V_B^p(\mathcal{H}^{|\psi\rangle} \otimes \mathcal{H}^{|\phi\rangle}))$'s. In the forth equality, we used the fact that the action of $V_B^p$ on $\mathcal{H}^{|\psi\rangle}$ is trivial. In the fifth equality, we used the fact that the action of $V_{\bar{B}}^m$ on $\mathcal{H}^{|\phi\rangle}$ is trivial. In the sixth equality, we expand $V_{\bar{B}}^m \mathcal{H}^{|\phi\rangle}$ in terms of $\Delta_q(V_{\bar{B}}^q \mathcal{H}^{|\phi\rangle})$'s. The seventh equality holds, because $\Delta_p(V_B^p \mathcal{H}^{|\psi\rangle}) \otimes \Delta_q(V_{\bar{B}}^q \mathcal{H}^{|\phi\rangle})$ with different values of $p$ and $q$ are orthogonal to each other. In the eighth line, we used Eq. (A.2).

Let $\Delta_k$ act on the left side and the last line of Eq. (A.9), we get

$$\Delta_k W_k^{|\psi\rangle|\phi\rangle} = \bigoplus_{l+m=k} \Delta_l W_l^{|\psi\rangle} \otimes \Delta_m W_m^{|\phi\rangle} . \tag{A.10}$$

Then compute the dimension

$$
\begin{aligned}
|\Delta_k W_k^{|\psi\rangle|\phi\rangle}| &= \sum_{l+m=k} |\Delta_l W_l^{|\psi\rangle} \otimes \Delta_m W_m^{|\phi\rangle}| \\
&= \sum_{l+m=k} |\Delta_l W_l^{|\psi\rangle}| |\Delta_m W_m^{|\phi\rangle}| .
\end{aligned}
\tag{A.11}
$$

Remind that these are coefficients of entanglement polynomials

$$
\begin{aligned}
f(|\psi\rangle)f(|\phi\rangle) &= \left(\sum_{l=0}^{N_{|\psi\rangle}} |\Delta_l W_l^{|\psi\rangle}| x^l\right)\left(\sum_{m=0}^{N_{|\phi\rangle}} |\Delta_m W_m^{|\phi\rangle}| x^m\right) \\
&= \sum_{k}^{N_{|\psi\rangle}+N_{|\phi\rangle}} x^k \sum_{l+m=k} |\Delta_l W_l^{|\psi\rangle}||\Delta_m W_m^{|\phi\rangle}| \\
&= \sum_{k}^{N_{|\psi\rangle}+N_{|\phi\rangle}} |\Delta_k W_k^{|\psi\rangle|\phi\rangle}||x^k \\
&= \sum_{k}^{N_{|\psi\rangle\otimes|\phi\rangle}} |\Delta_k W_k^{|\psi\rangle|\phi\rangle}| x^k \\
&= f(|\psi\rangle \otimes |\phi\rangle).
\end{aligned}
\tag{A.12}
$$

In the third equality, Eq. (A.11) is used. In the forth equality, we used $N_{|\psi\rangle\otimes|\phi\rangle} = N_{|\psi\rangle} + N_{|\phi\rangle}$. Thus Eq. (10) is proven. Note that if Eq. (10) is assumed, then we can derive Eq. (A.11) by comparing the coefficients of the second line and that of the forth line of Eq. (A.12). So Eq. (A.11) is equivalent to Eq. (10).

# B  Size additivity and polynomial factorization

Eq. (10) can be interpreted as a stronger form of the additivity of the state-dependent operator size. To illustrate it, we first briefly review several related notions.

## B.1  State-dependent operator size

Given a state $|\psi\rangle$, we can costruct a series of subspaces $\Delta W_k$ defined by Eq. (6). As mentioned in Section 2, states in $\Delta W_k$ cannot be generated by $k-1$-local operators acting on $|\psi\rangle$ but can be generated by $k$-local operators. For this reason, if an operator $\mathcal{O}$ transforms $|\psi\rangle$ into a state in $\Delta W_k$, then we are tempted to say that there are $k$ qudits are changed by $\mathcal{O}$ on average. This is not a rigorous statement at this stage, but it is an appropriate interpretation of $\Delta W_k$. In fact, this interpretation is my original motivation to design the definition of the state-dependent operator size [11].

Denote the projector onto $W_k$ as $P_k$. According to Eq. (6), the projector onto $\Delta W_k$ is given by

$$
\Delta P_k = P_k - P_{k-1}.
\tag{B.1}
$$

Suppose $|\psi\rangle$ is transformed into $|\phi\rangle$. According to the above interpretation, if $|\phi\rangle \in \Delta W_k$ for some $k$, we say that $k$ qudits are changed on average. In other words, there should be a size operator such that $\Delta W_k$ is its eigenspace with eigenvalue equal to $k$, i.e.

$$
n_{|\psi\rangle} = \sum_{k=0}^{N} k \Delta P_k.
\tag{B.2}
$$

This is the size operator mentioned in Eq. (1). The subscript of $n_{|\psi\rangle}$ indicates the dependence on $|\psi\rangle$.[2] $N$ is the number of all qudits. For a general state $|\phi\rangle$, the expectation value of the

---

[2]The range of $k$ starting from 0 in Eq. (B.2) is chosen for later convenience. One may notice that the subscript of $N_{|\psi\rangle}$ is omitted in Eq. (B.2). This does not contradict with Eq. (6). According to Eq. (B.1), $\Delta P_k = 0$ for $k > N_{|\psi\rangle}$. This subscript will also be omitted in the following formulas for the same reason.

number of changed qudits is

$$\mathcal{S}(|\phi\rangle,|\psi\rangle) = \langle\phi|n_{|\psi\rangle}|\phi\rangle\,. \tag{B.3}$$

We can call it the size of $|\phi\rangle$ relative to $|\psi\rangle$. Note that if we replace $|\phi\rangle$ by $\mathcal{O}|\psi\rangle/\langle\psi|\mathcal{O}^\dagger\mathcal{O}|\psi\rangle^{1/2}$, then the state-dependent operator size mentioned in Eq. (1) is recovered.

## B.2 Size additivity

Since the relative size defined in Eq. (B.3) is interpreted as the average number of changed qudits, if there are two independent systems $B_1$ and $B_2$, then the relative size should satisfy the additivity

$$\mathcal{S}(|\phi_1\rangle|\phi_2\rangle,|\psi_1\rangle|\psi_2\rangle) = \mathcal{S}(|\phi_1\rangle,|\psi_1\rangle) + \mathcal{S}(|\phi_2\rangle,|\psi_2\rangle)\,. \tag{B.4}$$

$|\phi_i\rangle, |\psi_i\rangle \in \mathcal{H}_{B_i}$ with $i = 1, 2$. $|\phi_1\rangle$ and $|\phi_2\rangle$ are arbitrary, so Eq. (B.4) is equivalent to

$$n_{|\psi_1\rangle|\psi_2\rangle} = n_{|\psi_1\rangle} + n_{|\psi_2\rangle}\,. \tag{B.5}$$

Before proving it, let me illustrate a stronger condition that should hold. Using Eq. (B.2) and Eq. (B.3), the relative size can be expressed as

$$\mathcal{S}(|\phi_i\rangle,|\psi_i\rangle) = \sum_{k=0}^{N} k\langle\phi_i|\Delta P_k^{|\psi_i\rangle}|\phi_i\rangle\,. \tag{B.6}$$

The superscript of $\Delta P_k^{|\psi_i\rangle}$ indicates its dependence on $|\psi_i\rangle$. $\langle\phi_i|\Delta P_k^{|\psi_i\rangle}|\phi_i\rangle$ can be interpreted as the probability that $k$ qudits are changed on average. We denote it as $\mathrm{Pr}_i(k)$ for later convenience,

$$\mathrm{Pr}_i(k) \equiv \langle\phi_i|\Delta P_k^{|\psi_i\rangle}|\phi_i\rangle\,. \tag{B.7}$$

We further denote the probability that $k_1$ qudits of $B_1$ and $k_2$ qudits of $B_2$ changed on average as $\mathrm{Pr}(k_1, k_2)$. Since we assume $B_1$ and $B_2$ are independent, we have

$$\mathrm{Pr}(k_1, k_2) = \mathrm{Pr}_1(k_1)\mathrm{Pr}_2(k_2)\,. \tag{B.8}$$

Then the probability, $\mathrm{Pr}(k_1 + k_2 = k)$, that the total number of changed qudits of $B_1$ and $B_2$ is equal to $k$ should be given by

$$\mathrm{Pr}(k_1 + k_2 = k) = \sum_{k_1+k_2=k} \mathrm{Pr}(k_1, k_2) = \sum_{k_1+k_2=k} \mathrm{Pr}_1(k_1)\mathrm{Pr}_2(k_2)\,. \tag{B.9}$$

Using Eq. (B.7), we get

$$\langle\phi_1|\langle\phi_2|\Delta P_k^{|\psi_1\rangle|\psi_2\rangle}|\phi_1\rangle|\phi_2\rangle = \sum_{k_1+k_2=k} \langle\phi_1|\Delta P_{k_1}^{|\psi_1\rangle}|\phi_1\rangle\langle\phi_2|\Delta P_{k_2}^{|\psi_2\rangle}|\phi_2\rangle\,. \tag{B.10}$$

Since $|\phi_1\rangle$ and $|\phi_2\rangle$ are arbitrary, we get

$$\Delta P_k^{|\psi_1\rangle|\psi_2\rangle} = \sum_{k_1+k_2=k} \Delta P_{k_1}^{|\psi_1\rangle}\Delta P_{k_2}^{|\psi_2\rangle}\,. \tag{B.11}$$

I should emphasize that, until now, we have not proven or derived Eq. (B.4) or Eq. (B.11). Instead, we are discussing what properties should hold for a proper definition of relative size.

To prove them, note that $\Delta P_k$ is the projector onto $\Delta W_k$, so Eq. (B.11) is equivalent to Eq. (A.10) which is proven in Section A. Eq. (B.4) can be derived from Eq. (B.11). To see it, note that

$$
\begin{aligned}
\mathcal{S}(|\phi_1\rangle|\phi_2\rangle, |\psi_1\rangle|\psi_2\rangle) &= \sum_{k=0}^{|B_1|+|B_2|} k \langle\phi_1|\langle\phi_2|\Delta P_k^{|\psi_1\rangle|\psi_2\rangle}|\phi_1\rangle|\phi_2\rangle \\
&= \sum_{k=0}^{|B_1|+|B_2|} k \sum_{k_1+k_2=k} \langle\phi_1|\Delta P_{k_1}^{|\psi_1\rangle}|\phi_1\rangle\langle\phi_2|\Delta P_{k_2}^{|\psi_2\rangle}|\phi_2\rangle \\
&= \sum_{k=0}^{|B_1|+|B_2|} \sum_{k_1+k_2=k} (k_1+k_2)\langle\phi_1|\Delta P_{k_1}^{|\psi_1\rangle}|\phi_1\rangle\langle\phi_2|\Delta P_{k_2}^{|\psi_2\rangle}|\phi_2\rangle \\
&= \sum_{k_1=0}^{|B_1|}\sum_{k_2=0}^{|B_2|} (k_1+k_2)\langle\phi_1|\Delta P_{k_1}^{|\psi_1\rangle}|\phi_1\rangle\langle\phi_2|\Delta P_{k_2}^{|\psi_2\rangle}|\phi_2\rangle \\
&= \sum_{k_1=0}^{|B_1|}\sum_{k_2=0}^{|B_2|} k_1\langle\phi_1|\Delta P_{k_1}^{|\psi_1\rangle}|\phi_1\rangle\langle\phi_2|\Delta P_{k_2}^{|\psi_2\rangle}|\phi_2\rangle \\
&\quad + \sum_{k_2=0}^{|B_2|}\sum_{k_1=0}^{|B_1|} k_2\langle\phi_1|\Delta P_{k_1}^{|\psi_1\rangle}|\phi_1\rangle\langle\phi_2|\Delta P_{k_2}^{|\psi_2\rangle}|\phi_2\rangle \\
&= \sum_{k_1=0}^{|B_1|} k_1\langle\phi_1|\Delta P_{k_1}^{|\psi_1\rangle}|\phi_1\rangle \sum_{k_2=0}^{|B_2|}\langle\phi_2|\Delta P_{k_2}^{|\psi_2\rangle}|\phi_2\rangle \\
&\quad + \sum_{k_2=0}^{|B_2|} k_2\langle\phi_2|\Delta P_{k_2}^{|\psi_2\rangle}|\phi_2\rangle \sum_{k_1=0}^{|B_1|}\langle\phi_1|\Delta P_{k_1}^{|\psi_1\rangle}|\phi_1\rangle \\
&= \sum_{k_1=0}^{|B_1|} k_1\langle\phi_1|\Delta P_{k_1}^{|\psi_1\rangle}|\phi_1\rangle + \sum_{k_2=0}^{|B_2|} k_2\langle\phi_2|\Delta P_{k_2}^{|\psi_2\rangle}|\phi_2\rangle \\
&= \mathcal{S}(|\phi_1\rangle, |\psi_1\rangle) + \mathcal{S}(|\phi_2\rangle, |\psi_2\rangle).
\end{aligned}
\tag{B.12}
$$

In the second equality, we used Eq. (B.10). $|B_i|$ is the number of qudits in $B_i$. So Eq. (B.11) is a stronger condition than Eq. (B.4). For this reason, we call Eq. (B.11) the strong additivity. Remind that Eq. (B.11) is equivalent to Eq. (A.10) and Eq. (A.10) implies Eq. (A.11) which is equivalent to Eq. (10). Thus, the strong additivity implies Eq. (10) and the size additivity Eq. (B.4).

Note that the entanglement polynomial is equal to the trace[3]

$$
f(|\psi\rangle) = \mathrm{tr}\left(\sum_{k=0}^{N} x^k \Delta P_k^{|\psi\rangle}\right).
\tag{B.13}
$$

We define an operator

$$
F^{|\psi\rangle} \equiv \sum_{k=0}^{N} x^k \Delta P_k^{|\psi\rangle}.
\tag{B.14}
$$

Then Eq. (B.11) is equivalent to

$$
F_{|\psi_1\rangle|\psi_2\rangle} = F_{|\psi_1\rangle} F_{|\psi_2\rangle}.
\tag{B.15}
$$

---

[3]The subscript of $N_{|\psi\rangle}$ is omitted for the same reason mentioned in footnote 2.

Eq. (10) can be derived by taking the trace. The size operator is related to $F$ by

$$n_{|\psi\rangle} = \partial_x F_{|\psi\rangle}|_{x=1} . \tag{B.16}$$

Using Eq. (B.15), we get

$$
\begin{aligned}
n_{|\psi_1\rangle|\psi_2\rangle} &= \partial_x (F_{|\psi_1\rangle|\psi_2\rangle})|_{x=1} \\
&= [(\partial_x F_{|\psi_1\rangle}) F_{|\psi_2\rangle}]|_{x=1} + [F_{|\psi_1\rangle}(\partial_x F_{|\psi_2\rangle})]|_{x=1} \\
&= \partial_x F_{|\psi_1\rangle}|_{x=1} + \partial_x F_{|\psi_2\rangle}|_{x=1} \\
&= n_{|\psi_1\rangle} + n_{|\psi_2\rangle} .
\end{aligned}
\tag{B.17}
$$

In the third equality, we used the fact that $F_{|\psi\rangle}|_{x=1}$ is equal to the identity. This is a simpler derivation of the size additivity.

## C  Proof of theorem 3

Note that

$$
\begin{aligned}
|V^k \mathcal{H}^{g|\psi\rangle}| &= |V^k g \mathcal{H}^{|\psi\rangle}| \\
&= |g g^{-1} V^k g \mathcal{H}^{|\psi\rangle}| \\
&= |g^{-1} V^k g \mathcal{H}^{|\psi\rangle}| \\
&= |V^k \mathcal{H}^{|\psi\rangle}| .
\end{aligned}
\tag{C.1}
$$

In the first equality, we defined the operator action on the Hilbert space

$$gW \equiv \mathrm{Span}\left\{ g|\psi\rangle \,\middle|\, |\psi\rangle \in W \right\} . \tag{C.2}$$

In the third equality, we used the fact that an invertible operator induces an automorphism on the vector space. Eq. (6) implies

$$|\Delta W_k| = |V^k \mathcal{H}^{|\psi\rangle}| - |V^{k-1} \mathcal{H}^{|\psi\rangle}| . \tag{C.3}$$

Note that

$$g^{-1} V g = V \Longleftrightarrow \forall\, k \in \mathbb{N}, \qquad g^{-1} V^k g = V^k . \tag{C.4}$$

Collecting Eq. (7), Eq. (C.1) and Eq. (C.3), Theorem 3 is proven.

## D  Proof of $W_k = \mathrm{Im}(\rho_k)$

In this section, we prove Eq. (31). We first prove two lemmas.

**Lemma 1** *Suppose that $\rho, \sigma \in End(W)$ are positive semi-definite operators and denote their image by $Im(\rho)$ and $Im(\sigma)$ respectively. We have*

$$\mathrm{Im}(\rho + \sigma) = \mathrm{Im}(\rho) + \mathrm{Im}(\sigma) . \tag{D.1}$$

**Proof 1** *By definition, a positive semi-definite operator $\mathcal{O}$ satisfies*

$$\langle\psi|\mathcal{O}|\psi\rangle = 0 \iff |\psi\rangle \in Ker(\mathcal{O}). \tag{D.2}$$

*$Ker(\mathcal{O})$ is the kernel of $\mathcal{O}$. Notice that*

$$\langle\psi|\rho+\sigma|\psi\rangle \iff \langle\psi|\rho|\psi\rangle = 0, \quad \langle\psi|\sigma|\psi\rangle = 0. \tag{D.3}$$

*Thus their kernel satisfy*

$$Ker(\rho+\sigma) = Ker(\rho) \cap Ker(\sigma). \tag{D.4}$$

*Since the kernel of a hermitian operator is the orthogonal complement of its image, Eq. (D.1) is proven.*

Remind that, given a state $\rho$, the reduced density matrix $\rho_{\bar{B}} \in \mathcal{A}_{\bar{B}}$ is defined by

$$tr(\mathcal{O}\rho_{\bar{B}}) = tr(\mathcal{O}\rho), \quad \forall\, \mathcal{O} \in \mathcal{A}_{\bar{B}}. \tag{D.5}$$

**Lemma 2** *Suppose $U \in \mathcal{A}_B$ is unitary and $dU$ is the normalized Haar measure. Given a state $\rho$, we have*

$$\int_{U \in \mathcal{A}_B} U\rho U^\dagger dU = \rho_{\bar{B}}. \tag{D.6}$$

**Proof 2** *Given an arbitrary operator $\mathcal{O} \in \mathcal{A}_{\bar{B}}$, we have*

$$\begin{aligned}
tr(\mathcal{O}\int_{U \in \mathcal{A}_B} U\rho U^\dagger dU) &= \int_{U \in \mathcal{A}_B} tr(\mathcal{O}U\rho U^\dagger)dU \\
&= \int_{U \in \mathcal{A}_B} tr(U\mathcal{O}\rho U^\dagger)dU \\
&= \int_{U \in \mathcal{A}_B} tr(U^\dagger U\mathcal{O}\rho)dU \\
&= tr(\mathcal{O}\rho)\int_{U \in \mathcal{A}_B} dU \\
&= tr(\mathcal{O}\rho).
\end{aligned} \tag{D.7}$$

*Since operators in $\mathcal{A}_B$ and $\mathcal{A}_{\bar{B}}$ commute, we have the second equality. In the last equality, we used the normalization*

$$\int_{U \in \mathcal{A}_B} dU = 1. \tag{D.8}$$

*Now we only need to prove $\int_{U \in \mathcal{A}_B} U\rho U^\dagger dU \in \mathcal{A}_{\bar{B}}$. Suppose $W$ is a unitary operator in $\mathcal{A}_B$.[4]*

---

[4]This "$W$" should not be confused with the symbol of the $k$-local subspace.

*Denote $WU$ by $U_W$, so $U = W^\dagger U_W$. Then we have*

$$
\begin{aligned}
W \int_{U \in \mathcal{A}_B} U \rho U^\dagger dU &= \int_{U \in \mathcal{A}_B} U_W \rho U^\dagger dU \\
&= \int_{U \in \mathcal{A}_B} U_W \rho (W^\dagger U_W)^\dagger d(W^\dagger U_W) \\
&= \int_{U \in \mathcal{A}_B} U_W \rho U_W^\dagger W d(W^\dagger U_W) \\
&= \left( \int_{U \in \mathcal{A}_B} U_W \rho U_W^\dagger dU_W \right) W \\
&= \left( \int_{U \in \mathcal{A}_B} U \rho U^\dagger dU \right) W \, .
\end{aligned}
\tag{D.9}
$$

*Since any operator can be expressed as a linear combination of unitary operators, we have*

$$
\left[ \mathcal{O}, \int_{U \in \mathcal{A}_B} U \rho U^\dagger dU \right] = 0, \quad \forall \, \mathcal{O} \in \mathcal{A}_B \, .
\tag{D.10}
$$

*Since the Hilbert space is factorized, we conclude that $\int_{U \in \mathcal{A}_B} U \rho U^\dagger dU$ is in $\mathcal{A}_{\bar{B}}$.*

Combine the above two lemmas, we can prove the following identity for a pure state $\rho$,

$$
W_k = \text{Im}(\rho_k) \, .
\tag{D.11}
$$

To prove it, we note that

$$
\begin{aligned}
\text{Im}(\rho_k) &= \text{Im}\left( \sum_{|B|=k} \rho_{\bar{B}} \right) \\
&= \sum_{|B|=k} \text{Im}\left( \rho_{\bar{B}} \right) \\
&= \sum_{|B|=k} \text{Im}\left( \int_{U \in \mathcal{A}_B} U \rho U^\dagger dU \right) \\
&= \sum_{|B|=k} \sum_{U \in \mathcal{A}_B} \text{Im}\left( U \rho U^\dagger \right) \\
&= \sum_{|B|=k} \text{Span}\{ U |\psi\rangle \, \big| \, U \in \mathcal{A}_B \} \\
&= \sum_{|B|=k} \mathcal{A}_B \mathcal{H}^{|\psi\rangle} \\
&= V^k \mathcal{H}^{|\psi\rangle} \\
&= W_k \, .
\end{aligned}
\tag{D.12}
$$

In the second, fourth, and fifth equality, we used Lemma 1. In the third equality, we used Lemma 2. Since any operator can be expanded by unitary operators, we have the sixth equality. The other equalities hold by definition. Eq. (D.11) implies Eq. (31) immediately.

There are different ways to define renormalized states. Remind that renormalized states defined in Eq. (26) correspond to the uniform probability distribution. Now we consider general distributions. Denote the probability distribution of lost part $B$ as $p(B)$. Then for a given

state $\rho$, the average of the expectation value of $\mathcal{O}$ in the noisy laboratory is

$$\overline{\langle\mathcal{O}\rangle_{\text{noise}}} = \sum_B p(B)\text{tr}(\mathcal{O}\rho_{\tilde{B}})\,. \tag{D.13}$$

Define the renormalized state

$$\rho_k \equiv \sum_{|B|=k} p(B)\rho_{\tilde{B}}\,. \tag{D.14}$$

If the measure of lost part is fixed to be $k$, then we have

$$\overline{\langle\mathcal{O}\rangle_{\text{noise}}} = \text{tr}(\mathcal{O}\rho_k)\,. \tag{D.15}$$

Using Lemma 1, note that

$$\text{Im}(\rho_k) = \sum_{|B|=k} \text{Im}[p(B)\rho_{\tilde{B}}] = \sum_{|B|=k} \text{Im}(\rho_{\tilde{B}})\,, \tag{D.16}$$

for positive definite $p(B)$. Thus, the derivation of Eq. (D.12) applies to the case of any positive definite distribution.

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
