# Peer review of "Entanglement Classification via Operator Size"

_SciPost Physics, doi:SciPost Phys. Core 6, 063 (2023)_

## Round 6 · Referee Report · Anonymous (Referee 2) · 2022-12-28

Report

The author constructs a polynomial that can classify the entanglement structure of pure states. Intuitively, this polynomial associated to a pure state $|\psi \rangle$ is essentially given by how many orthogonal states can be span by applying a $k$-local operator to that state $|\psi\rangle$. This map from states to polynomials is a homomorphism (and an isomorphism by a proper quotient), where the tensor product of quantum states induces the product of the corresponding polynomial. Then the author proves that the entanglement polynomial is SLOCC invariant. These are the main results of this paper. Later, the author provides a way to calculate the entanglement polynomial and apply it to three-qubit states.

This paper is well-written and the result is interesting and important. The classification of entanglement of quantum states, in particular, with many qubits, is one of the most important problems in quantum information science. The construction of entanglement polynomials may help lead to a better understanding of many-qubit quantum states. Therefore, I recommend its publication in SciPost Physics with a small comment: It would be nice to have some more discussions of the meaning of the entanglement polynomial besides that it serves as a SLOCC invariant.
  • validity: -
  • significance: -
  • originality: -
  • clarity: -
  • formatting: -
  • grammar: -

Author:  Qi-Feng Wu  on 2023-01-09  [id 3219]

(in reply to Report 1 on 2022-12-28)

Thanks for your helpful report.

Regarding the meaning of the entanglement polynomial besides it being a SLOCC invariant, it is also closely related to quantum chaos. As mentioned in the introduction of the paper, the entanglement polynomial is motivated by a generalized operator size that measures quantum scrambling. In fact, the entanglement polynomial is a special case of the state-dependent operator size. It is explained in Appendix B. So the entanglement polynomial should play a role in quantum chaos. Operator size in the SYK model is shown to be dual to the SL(2, R) generators of the JT gravity. So the entanglement polynomial should also be related to symmetries in the JT gravity.

---

## Round 6 · Referee Report · Anonymous (Referee 3) · 2023-2-2

Report

In this paper, the author defined the entanglement polynomial to characterize the multipartite entanglement. The author also proved that two states share the same entanglement polynomial if they are equivalent under the stochastic local operations and classical communication. Furthermore, the authors constructed the renormalized states and demonstrated how to evaluate the entanglement polynomial practically. This paper can be potentially published on SciPost after making the following revisions:

(1) It is difficult to follow the physics in this paper. The author needs to rewrite the introduction to make it readable. For instance, at the beginning of the introduction, the author discusses the operator dynamics in chaotic system which is essential in understanding the information spreading/scrambling (Ref. [1-4]). However, I do not find much discussion about the connection between the entanglement polynomial and operator dynamics.

(2) In the introduction, the author states that: “there is an ambiguity that operators of different sizes can act on a given state in the same way, which is a common phenomenon in the stabilizer formalism”. What does this mean? Does the entanglement polynomial can resolve this ambiguity claimed by the author? How to use the entanglement polynomial to characterize the multipartite entanglement in the stabilizer state?

(3) In the past few years, there have been a lot of progress in understanding the entanglement dynamics and operator dynamics in various chaotic/integrable systems. Does the entanglement polynomial provide new insight in understanding the multipartite entanglement structure. Also at the beginning of the abstract, the author claimed that “a systematic approach to classifying multipartite entanglement is developed”. However, I do not see investigation about the classification of multipartite entanglement in different systems.

Requested changes

  1. Rewrite the introduction (see the report)
  2. Explain how to use entanglement polynomial to provide new perspective beyond the current work on operator dynamics and entanglement dynamics.
  3. How to classify the multipartite entanglement in various systems by using the tool developed by the authors?

---

## Round 7 · Referee Report · Anonymous (Referee 4) · 2023-7-12

Report

The current version looks good and can be published on Scipost.

---

## Editorial Decision

published